# CLUES A Comprehensive Workflow for Integrating Geospatial Data in Biomedical Research

Marcel Jentsch[1,48], Elli Polemiti [1,2,48], Paul Renner [3,4,48], Sören Hese[3], Kerstin Schepanski [4], Roland Eils [1,2], Andre Marquand [5,49], Sven Twardziok [1,6,49] ✉, Gunter Schumann[6,7,8,9,49] & On behalf of the environ-MENTAL consortium*

Environmental exposures play a critical role in shaping physical and mental health, yet integrating such data into biomedical research remains technically complex and fragmented. The EnvironMENTAL Climate, Urbanicity, Environment and Society (CLUES) framework is an open-source, end-to-end workflow for generating individual-level environmental exposure data. CLUES automates the selection and download of open-access geospatial datasets, standardises spatial and temporal formats, and maps projections, and links resulting environmental variables to individual-level biomedical data, requiring no prior expertise in geospatial data. CLUES covers key environmental domains, including urban and natural space, climate and weather extremes, air pollution, and regional socioeconomic conditions. Designed for extensibility and cross-cohort applicability, it enables multidimensional exposure mapping across global settings and adheres to FAIR (Findability, Accessibility, Interoperability and Reusability) and privacy-compliant data protection principles. In this work, we present the CLUES framework and evaluate its scalability, computational performance, and reproducibility for large-scale biomedical research.

Environmental exposures are important determinants of human health, influencing a wide range of physical and psychological outcomes. Integrating environmental data with clinical and genomic information enables a more comprehensive understanding of health and disease. For example, in a large UK population-based study, an urban profile characterised by social deprivation, air pollution, dense street network and land-use was associated with affective symptoms. These associations were mediated by brain volume differences related

[1]Berlin Institute of Health at Charité – Universitätsmedizin Berlin, Center of Digital Health, Berlin, Germany. [2]Health Data Science Unit, Bioquant, Medical Faculty, University of Heidelberg, Im Neuenheimer Feld 267, Heidelberg, Germany. [3]Department of Earth Observation, Institute of Geography, Friedrich Schiller University Jena, Leutragraben 1, Jena, Germany. [4]Freie Universität Berlin, Department of Earth Sciences, Institute of Meteorology, Carl-Heinrich-Becker Weg 6-10, Berlin, Germany. [5]Radboud University Medical Center, Geert Grooteplein Zuid 10, Nijmegen GA, Netherlands. [6]German Centre for Mental Health (DZPG), Berlin-Potsdam site, Berlin, Germany. [7]Centre for Population Neuroscience and Stratified Medicine (PONS), Department of Psychiatry and Neuroscience, Charité – Universitätsmedizin Berlin, Berlin, Germany. [8]Centre for Population Neuroscience and Precision Medicine (PONS), Institute for Science and Technology of Brain-inspired Intelligence (ISTBI), Fudan University, Shanghai, PR China. [9]National Center for Neurological Disorders, Huashan Hospital, Fudan University, Shanghai, PR China. [48]These authors contributed equally: Marcel Jentsch, Elli Polemiti, Paul Renner. [49]These authors jointly supervised this work: Andre Marquand, Sven Twardziok, Gunter Schumann. *A list of authors and their affiliations appears at the end of the paper. ✉e-mail: sven.twardziok@bih-charite.de

to reward processing and moderated by stress response genes[1]. Beyond urbanisation, exposures such as rising temperatures, extreme weather events, and declining air quality are associated with an increased risk of cardiovascular and respiratory conditions, as well as the spread and severity of infectious diseases[2–7].

Large-scale cohort and clinical studies often collect detailed biological and clinical data, along with time and location information that can be used for geospatial linkage to external environmental datasets. To investigate the role of environmental exposures in shaping mental and physical health, researchers need access to geospatial data that combine spatial coordinates with rich environmental attributes and temporal information to capture dynamic exposures[8]. Integrating such contextual information to individual-level data supports systems-level analyses, integrating genomics, metabolomics, neuroimaging and other modalities.

Although the growing availability of open-access environmental data through platforms such as the Europe's Copernicus Programme, the National Aeronautics and Space Administration (NASA), or the German Aerospace Centre (DLR) offers opportunities for environmental research, it also introduces significant challenges. These include large data volumes of high spatial and temporal resolution data, differences in data access protocols, fragmentation across repositories, and variations in how data is formatted and structured. Additional obstacles involve spatial and temporal misalignment, such as differences in map projections, spatial coverage and time periods, as well as the need for specialised expertise to access, process, and integrate datasets from heterogeneous sources. Recent Earth Science discussions have further highlighted that most existing environmental data products are not sufficiently aligned with user needs, particularly in terms of FAIR (Findable, Accessible, Interoperable and Reusable) principles, and often lack long-term support and usability across disciplines[9]. These challenges prevent the routine inclusion of environmental exposures in biomedical research, despite their relevance.

To address these gaps, we developed the EnvironMENTAL Climate, Urbanicity, Environment and Society (CLUES) framework. CLUES is an open-source infrastructure that automates the generation of a curated database and provides tools to enable researchers across diverse disciplines to map large-scale environmental data to their datasets. It performs the full pipeline, including the selection and download of open-access global environmental data from a curated catalogue, harmonization into a unified format, and linkage to geospatial health datasets. The result is a participant-specific environtype[10], a composite descriptor of environmental exposures, incorporating spatial, temporal, and thematic dimensions. Early studies within the environMENTAL consortium (www.environmental-project.org) demonstrated the feasibility of linking satellite-derived remote sensing data, initially focused on a single environmental dimension, with large-scale neuroimaging cohorts, including the European IMAGEN study[11], the China Imaging Genetics (CHIMGEN) project[12], and the US Adolescent Brain Cognitive Development (ABCD) study[13]. However, these efforts relied on project-specific pipelines, limiting reproducibility, and the capacity to explore multidimensional environmental influences. CLUES overcomes these limitations through a unified and extensible infrastructure that supports a substantially broader and more diverse set of environmental data. It enables the integrated analysis of multiple environmental dimensions and their interplay in shaping health outcomes, while allowing researchers to tailor variable selection to their specific scientific questions. CLUES is currently being applied across several large-scale cohort studies, including the UK Biobank (UKBB)[14], IMAGEN[11], the German National Cohort (NAKO)[15], the German Health Interview and Examination Survey for Children and Adolescents (KiGGS)[16], and the Norwegian Mother, Father and Child Cohort Study (MoBa)[17].

CLUES is designed to provide a reproducible, scalable, and efficient workflow suitable for modern large-scale research. In line with this objective, we evaluated CLUES' scalability, computational performance, and reproducibility. The framework further enhances accessibility for users with limited expertise by including a set of Python scripts for linking environmental data to cohorts and notebooks for visualisation, quality checks, and exploratory analyses. Importantly, the framework can be deployed in secure environments compliant with data protection regulations such as the European General Data Protection Regulation (GDPR)[18]. Its modular structure enables the integration of new variables and the updating of the database as new timepoints and observations become available. By standardising formats, projections, and metadata, and by providing open, analysis-ready outputs, CLUES promotes adherence to the FAIR criteria, significantly lowering the technical barriers to environmental health research.

In this work, we demonstrate the application of CLUES and describe the structure and expected outputs of the framework, followed by an evaluation of its performance. CLUES delivers reproducible, scalable, and efficient performance across diverse environmental datasets, resolutions, and cohort settings. A detailed description of its configuration, data sources, and implementation is provided in the Methods, including a reproducible step-by-step guide.

## Results

CLUES supports integration of a wide range of environmental domains on (i) land use, including urbanicity[19,20], green spaces[21], water body structure[22] and other land cover layers[23]; (ii) topographical information[24]; (iii) atmospheric measurements[25,26], capturing temperature, precipitation, wind, humidity, and air pollutants; (iv) socioeconomic and demographic indicators, such as population density, income levels, employment status, education, and access to healthcare and transportation[27]. A full list of supported variables and data sources is available in the git documentation (https://bih-dmbs.github.io/CLUES/).

The CLUES workflow generates a harmonized environmental database, structured in configuration files that set execution and source/variable-depended parameters. Depending on the configuration, the workflow downloads the relevant files containing geospatial data from the primary sources and stores them in the local file system, in the data folder defined in the config.json. Each data source is stored in a dedicated folder, with subfolders corresponding to selected variables and time intervals. Each primary source has its own configuration file that governs how the data are organized and stored. Climate and atmospheric data are stored in separate files for each requested year. Although the database structure is configurable, standardised naming conventions and directory hierarchies enhance reproducibility and efficient data handling across projects.

### Individual-Level Linkage and Privacy Compliance

CLUES links individual-level data to environmental exposures using geographic coordinates as the required input (Fig. 1). Coordinates may be provided directly or obtained from address information using the geocoding script included in the online documentation. Routes (e.g., movement paths or GPS traces), which consist of coordinate sequences, can be linked through the standard workflow. A route-processing example notebook is also available online.

The framework is designed for secure, local deployment within institutional environments, ensuring that sensitive geolocation data are not transferred externally. In the IMAGEN study, for example, participant locations (accurate to 100 m) were linked to environmental variables using project-specific pseudo-identifiers. The resulting dataset, containing only derived exposures, was returned to the data provider for re-association with health data. All original geolocation information was deleted immediately after linkage. In one study center, where sharing geolocation data was not permitted, CLUES was

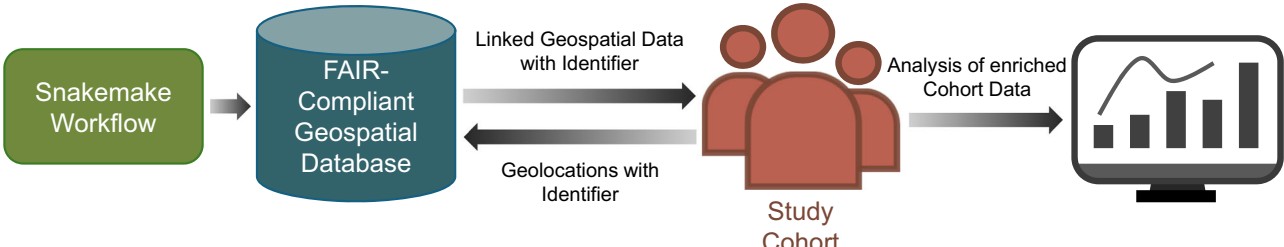

**Fig. 1 | Schematic illustration of the CLUES linkage workflow.** Participant location and time metadata are used to generate a personalized environmental profile ("environtype"). These enriched datasets can be used for statistical modeling, omics integration, and disease risk analysis. FAIR; Findability, Accessibility, Interoperability, Reusability.

deployed locally, allowing linkage to occur entirely within the institution's infrastructure.

In contexts where CLUES is implemented directly by authorised data holders, the use of pseudo-identifiers or coordinate obfuscation (e.g., rounding or masking) is not required. This flexibility supports data protection compliance while allowing integration of high-resolution environmental exposures into clinical, genomic, or neuroimaging studies.

## Performance, Reproducibility, and Interoperability

CLUES is designed for scalable, reproducible, and user-friendly deployment. To support interactive and collaborative use, we have deployed a database created using CLUES within a JupyterHub instance[28] with pre-configured Jupyter notebooks for, analysing, summarising and visualising the generated environmental data. This setup promotes hands-on learning, lowers the technical barrier for new users, and accelerates integration of CLUES into existing analytical pipelines. To access the JupyterHub workflows, visit https://jhub.bihealth.org/ and follow the instructions provided.

To demonstrate performance at scale, we applied CLUES to generate a national dataset for Norway, covering over 1.8 million square kilometers and spanning the years 2000 to 2024. The full environmental dataset that is based on the default source-configuration files, encompassing atmospheric, land-use, and topographical variables, required approximately 1 TB of storage and was generated in under four days.

Workflow execution time depends on multiple factors, including geographic extent, time range, and data source responsiveness. For example, downloading climate data for one year for the European region from Copernicus Climate Data Store took several minutes before the download started and approximately five hours for the downloading. Neighbourhood-based spatial metrics such as topographic smoothing depend on kernel size, i.e., larger kernels require more time. In our tests, generating a digital elevation model with 500 m kernel for Europe took around three hours.

Linking a dataset with 300,000 observations using the Norwegian dataset mentioned above took less than 2 hours. Additional performance and scalability benchmarks, including multi-country runtime comparisons and controlled raster-linkage evaluations, are provided in the Supplementary Information (SI 1).

## General Applicability Across Scientific Domains

The CLUES framework was developed to support research on environmental exposures and mental health. However, its modular and extensible design allows for broad applicability across diverse domains of biomedical and public health research, to explore a wide range of research questions in environmental epidemiology, exposomics, precision public health, and spatially resolved risk modelling. CLUES can be applied to multi-site consortia, policy evaluation studies, or urban and regional planning contexts and it is an ideal tool for comparative analyses across geographic regions or time periods. CLUES is not restricted to the application in health research as it can be adopted to other scientific realms as biology, sociology, or other empirical sciences where complex, multilevel data structures are common.

The workflow is not tied to any specific geographic region. Most default environmental datasets have global coverage, and researchers can tailor the environmental database to their research needs by defining the geographic extent, time window, and relevant environmental variables through editable configuration files. The framework's modularity also supports the integration of additional datasets when available, enabling seamless scaling across cohorts, regions, and research questions.

CLUES focuses on the upstream integration and linkage of environmental datasets, while leaving downstream statistical analysis entirely to the user. Environmental layers (e.g., GeoTIFF and netCDF files) can be used directly in standard GIS software (such as QGIS or ArcGIS) or in common analytical environments such as Python (e.g., xarray, rioxarray, rasterio, geopandas) and R (e.g., terra, sf, raster). The linkage step produces flat, analysis-ready exposure tables (CSV and JSON) that can be directly merged with user datasets and imported into any statistical software, including Python, R, or Stata. The linked exposure tables can be analysed using a wide range of methods, including area- or cluster-robust regression models, mixed-effects models, latent-variable models, or machine-learning approaches. To support both workflows, the online documentation includes example notebooks in Python. One set is demonstrating how to load, visualise, and explore generated geospatial layers, and another is illustrating how linked exposure tables can be analysed using a random-forest model on a synthetic dataset.

CLUES adheres to the FAIR principles, a set of guidelines aimed at improving the management and sharing of scientific data[29]. A comprehensive, open-source registry of supported variables and data sources ensures findability (https://bih-dmbs.github.io/CLUES/). Accessibility is enhanced through automated downloads from multiple sources, reducing the need for technical expertise and broadening the scope of available information. Interoperability is achieved through harmonisation of data types, coordinate reference systems, and bounding boxes (i.e., rectangular spatial limits that specify the geographic area of interest), allowing consistent use of data across tools and studies. Reusability is supported through the use of structured configuration files that enables reproducible database creation and consistent data processing across projects and teams[30].

Moreover, CLUES supports data handling with respect to applicable data protection regulations by enabling secure, local processing of sensitive geolocation data. This ensures that linkage between individual-level data and environmental exposures can be performed entirely within authorized institutional environments, eliminating the need for external data transfer and reducing privacy risk. Users may optionally apply additional privacy-preserving methods before linkage, such as adding small random offsets (jittering), aggregating or

snapping coordinates to grid cells, generating multiple plausible locations, or applying spatial smoothing, when required by local governance policies. These methods are not built into CLUES and should be applied externally. An example script illustrating coordinate jittering is provided in the online documentation.

## Discussion

The CLUES framework addresses a critical gap in geospatial data integration for biomedical and environmental research. By enabling automated access, harmonization, and linkage of diverse environmental exposures to individual-level health data, CLUES supports scalable, reproducible, and privacy-compliant enrichment of large population and clinical datasets. It facilitates cross-domain analyses that combine climate, urban, and socioeconomic data with biological layers such as genomics, neuroimaging, or health records. Designed for flexibility, CLUES is adaptable to a broad range of research contexts, ranging from public health and exposomics to urban planning, and ecology.

Unlike other platforms for environmental data access, which typically lack the flexibility to integrate multiple environmental domains, CLUES is specifically designed to generate customised, locally accessible geospatial datasets by combining data from a variety of primary sources. Other platforms do not aim for the same goal. They either specialise in a narrow environmental theme, or require substantial manual effort to add additional data or harmonise data across sources. Platforms, such as, Google Earth Engine, Microsoft Planetary Computer, or Copernicus Data Space Ecosystem, rely on cloud-based infrastructures and are dependent on Application Programming Interfaces (APIs), Web Map Services (WMSs), and other download tools. These tools, while powerful, typically demand significant geospatial expertise. Moreover, cloud-based services can pose data governance challenges, particularly in contexts where uploading geolocation data of participants is restricted by legal or institutional regulations.

In contrast, CLUES was designed to overcome these limitations. It provides an end-to-end, locally executable workflow that automates the download, harmonization, and integration of environmental data from diverse domains. Its modular structure and use of the Snakemake[31] workflow engine ensure reproducibility and scalability, supporting national-scale datasets and multi-decade timeframes with minimal manual intervention. The framework enables privacy-compliant data linkage within secure computing environments, without the need to expose identifiable information externally. In addition, CLUES is fully open-source, with pre-configured workflows, a documented variable catalogue, and user-friendly Jupyter notebooks that support researchers with limited technical expertise.

To our knowledge, no existing tool offers this combination of multi-domain environmental data integration, local privacy-compliant execution, reproducibility, and accessibility within a single, unified framework. CLUES was developed motivated by the practical challenges encountered during our research and within the environMENTAL consortium, where no existing solution met the demands of large-scale environmental data integration while respecting data protection regulations that come along with cohort data.

Some limitations of CLUES should be acknowledged. First, CLUES is dependent on external data providers. Changes in provider APIs, formats or availability can temporarily disrupt access or require adaptation. Therefore, CLUES will be actively maintained and updated to accommodate changes in data access protocols and ensure continued compatibility with evolving data infrastructures. As the data coverage may not always align with the desired temporal and geographic extents for specific studies, efforts were made to include global datasets with extended temporal resolution, enabling applicability across a wide range of geographic contexts and historical timeframes. While CLUES supports local database creation to comply

with privacy or institutional requirements, this setup may limit accessibility for external collaborators. This is not a limitation of the framework itself, but rather a reflection of data governance constraints. In such cases, high-speed secure file transfer solutions, such as RClone, can be used to share data products efficiently. Currently CLUES does not perform an internal quality control (QC) step. However, the framework relies on data that have undergone QC by the primary providers; thus, adhering to a baseline quality standard. CLUES currently supports only open-access datasets. While this ensures broad usability, it excludes some proprietary sources that may offer enhanced spatial or thematic resolution. Computational demands, such as processing time and memory consumption, may increase substantially when applying neighbourhood-based metrics with large kernel sizes, especially when working with high-resolution imagery or continental-scale data. Finally, for applications that require visual overlay or joint spatial analysis of raster layers from different projections (e.g., WGS84 and MODIS sinusoidal), users may need to reproject datasets to a common CRS. CLUES does not perform this reprojection by default, as it preserves datasets in their native projections to prevent unnecessary resampling and loss of spatial accuracy. General tutorials for reprojection of raster data are provided in the online documentation for users who require harmonised projections.

Despite these limitations, the CLUES framework provides a robust foundation for reproducible environmental data integration. Many of its constraints are either inherent to geospatial data ecosystems or mitigated through modular updates and the involvement of CLUES in ongoing and future research endeavours.

## Methods

### Geospatial Data Sources and Characteristics

CLUES incorporates geospatial data of a wide array of environmental conditions across spatial and temporal dimensions. The data can be grouped into the following four categories:

1. Urban and natural space: metrics related to urban development, build-up areas, vegetative cover, spatial arrangement of water bodies and nighttime lights, drought conditions, and topographic features, such as elevation, slope, and roughness.
2. Weather and climate: meteorological variables, comprising measurements of temperature, precipitation, wind, snow and cloud cover, sunshine duration, and extreme weather events[32,33].
3. Air quality: indicators of atmospheric pollution, including concentrations of gases such as nitrogen dioxide, sulphur dioxide, and ozone and measurements of various species of aerosols present in the atmosphere.
4. Socioeconomic data: demographic information such as population structure, migration patterns, and ageing; economic indicators including employment rates, economic performance, and regional GDP; and transport accessibility, transport networks, and mobility patterns.

The framework prioritises the inclusion of globally available datasets to support broad geographic applicability. However, region-specific data products with superior thematic detail or accessibility are also incorporated when appropriate. For example, for land cover data, CLUES includes both the Global Dynamic Land Cover dataset (Copernicus Global Land Service) and the CORINE (Coordination of Information on the Environment) Land Cover dataset. The latter, although specifically designed for Europe, provides a more detailed inventory of land cover, featuring 44 thematic classes that range from broad forested areas to individual vineyards.

The datasets used in CLUES vary in spatial and temporal resolution, as well as in geographic and temporal coverage, depending on their source and intended application. For instance, the Tree Cover Density dataset from Copernicus Land Monitoring Service is available for 2012, 2015 and annually from 2018 to 2021 reference years,

**Table 1 | Overview of data sources and services used by CLUES**

| Online Service | Product description |
|---|---|
| Copernicus (European Commission) | The European Union's Copernicus programme provides comprehensive and high-quality geospatial data through its constellation of Sentinel satellites and contributing missions. It delivers information across six thematic areas: land, ocean, atmosphere, climate change, emergency, and security. CLUES utilizes: the Copernicus Atmosphere Monitoring Service (CAMS), ERA5 reanalysis data, Dynamic Land Cover, CORINE Land Cover, and Digital Elevation Model[25,26]. |
| Earthdata (National Aeronautics and Space Administration (NASA)) | The Moderate Resolution Imaging Spectroradiometer (MODIS) is an instrument aboard NASA's Terra and Aqua satellites that collects data in 36 spectral channels with global coverage every 1-2days. CLUES utilizes the normalized difference vegetation index (NDVI) and the enhanced vegetation index (EVI). The two products characterize the global range of vegetation states and processes. Requires NASA Earthdata account registration[40]. |
| EOC (Earth Observation Center) Geoservices (German Aerospace Center (DLR) | DLR's EOC provides access to Earth observation data, including Atmosphere Coverage Service, Land Map Service, and Elevation Map Service. |
| Defense Meteorological Satellite Program (DMSP) and Visible Infrared Imaging Radiometer Suite (VIIRS) | This dataset, downloadable from Figshare, combines inter-calibrated DMSP observations with simulated DMSP-like VIIRS observations, ensuring continuity and comparability across the entire time series. Figshare is an online digital repository for researchers to store, share, and publish research outputs, including datasets, figures, and other scholarly materials[20]. |
| Hydrological data and maps based on Shuttle Elevation Derivatives at multiple Scales (HydroSHEDS) | HydroSHEDS is a comprehensive mapping product that provides high-resolution hydrographic information. Developed by the World Wildlife Fund (WWF) in collaboration with various scientific institutions, HydroSHEDS offers a suite of core data products, including digital elevation models, flow direction maps, and flow accumulation maps. CLUES utilizes the Global Lakes and Wetlands Database[22]. |
| The European Observation Network for Territorial Development and Cohesion (ESPON) | ESPON is an EU-funded program aimed at supporting policy development for territorial cohesion and harmonious development across Europe. ESPON provides a wealth of assets, including comparable information, evidence, analyses, and scenarios on territorial dynamics. ESPON is currently the only socioeconomic data source that is used by the CLUES, therefore socioeconomic data is only available for European countries[41]. |
| Drought Monitor (Spanish National Research Council) | The Standardised Precipitation-Evapotranspiration Index (SPEI) is a multiscale drought index that integrates both precipitation and potential evapotranspiration to assess drought conditions. The SPEI is designed to capture the impact of temperature variations on water demand, making it particularly useful for evaluating the effects of climate change on drought severity[42,43]. |
| WorldPop (*WorldPop Research Programme, University of Southampton*) | WorldPop provides high-resolution gridded population datasets derived from census and satellite data using geospatial modelling to map global human population distributions, demographics, and dynamics. CLUES currently integrates the WorldPop population count layer[44]. |

enabling periodic assessments and validation of satellite imagery. This supports high-resolution, reliable information on global tree cover density. In contrast, climate and atmospheric datasets typically feature a higher temporal resolution. The Copernicus ERA5 reanalysis dataset, for instance, provides hourly estimates of numerous atmospheric and land-surface variables from 1940 to the present by incorporating historical observational data in advanced modelling and assimilation systems. Thus, ERA5 delivers a spatially and temporally consistent dataset at a global scale with a spatial resolution of 0.25° x 0.25°.

Regional socioeconomic data integration presents additional challenges due to country-specific collection methods, definitions, and standards. For this reason, CLUES currently integrates data from the European Union's "Nomenclature des unités territoriales statistiques" (NUTS) hierarchical classification system[34], which standardizes territorial units across Europe allowing cross-border comparisons and regional policy assessments. While global datasets like the World Bank (https://data.worldbank.org/) and UNdata (https://data.un.org/) exist, their limited temporal and spatial resolution precludes their inclusion at this stage.

To effectively use the CLUES database for specific scientific inquiries, it is important to understand the characteristics and attributes of the available data. An overview of the features of each dataset is available on the CLUES GitHub repository (https://bih-dmbs.github.io/CLUES/dataCatalog/).

CLUES accesses geospatial data through heterogenous technical protocols and APIs. These include: (1) WMS[35] a standardised protocol established by the Open Geospatial Consortium (OGC), which delivers dynamic map images on parameters like geographic bounding box and coordinate reference system. Adhering to OGC standards ensures consistent and reliable access to geospatial data. (2) Climate Data Store (CDS) API[36], developed by the Copernicus Climate Change Service

(C3S), is a Python-based interface for accessing a vast repository of climate and atmospheric data. The CDS API requires a user authentication via a CDS account and API key and allows users to specify datasets, variables, time windows, and output format. (3) Copernicus Open Data (OData) Service[37] adhering to the OASIS (Organization for the Advancement of Structured Information Standards) standard, provides a RESTful interface for accessing and manipulating data within the Copernicus Data Space Ecosystem using standardised HTTP requests. The OASIS standard refers to a set of technical standards, developed by OASIS. RESTful refers to web services that follow the Representational State Transfer (REST) architectural style. Direct download links, which are available for several datasets, simplifying data acquisition for end users. These diverse access methods reflect the complexity of the geospatial data ecosystem, requiring researchers to navigate multiple standards, authentication protocols, and interfaces to effectively retrieve and integrate environmental data. By supporting a broad spectrum of data access protocols, CLUES enhances reproducibility and scalability of environmental data integration into research workflows.

Table 1 provides an overview of the services employed by CLUES to access different categories of geospatial data. For each service, the underlying technology and the thematic data types available for download are described.

**Workflow specifications**

The CLUES framework is developed using the Snakemake workflow management system and made publicly available as open-source software (Fig. 2). Its modular design allows for easy customization and regular updates. Configuration files define key parameters, such as selected variables, data storage locations, and the geographic and temporal extent, ensuring reproducible database creation. The Snakemake workflow coordinates dataset retrieval and processing, after

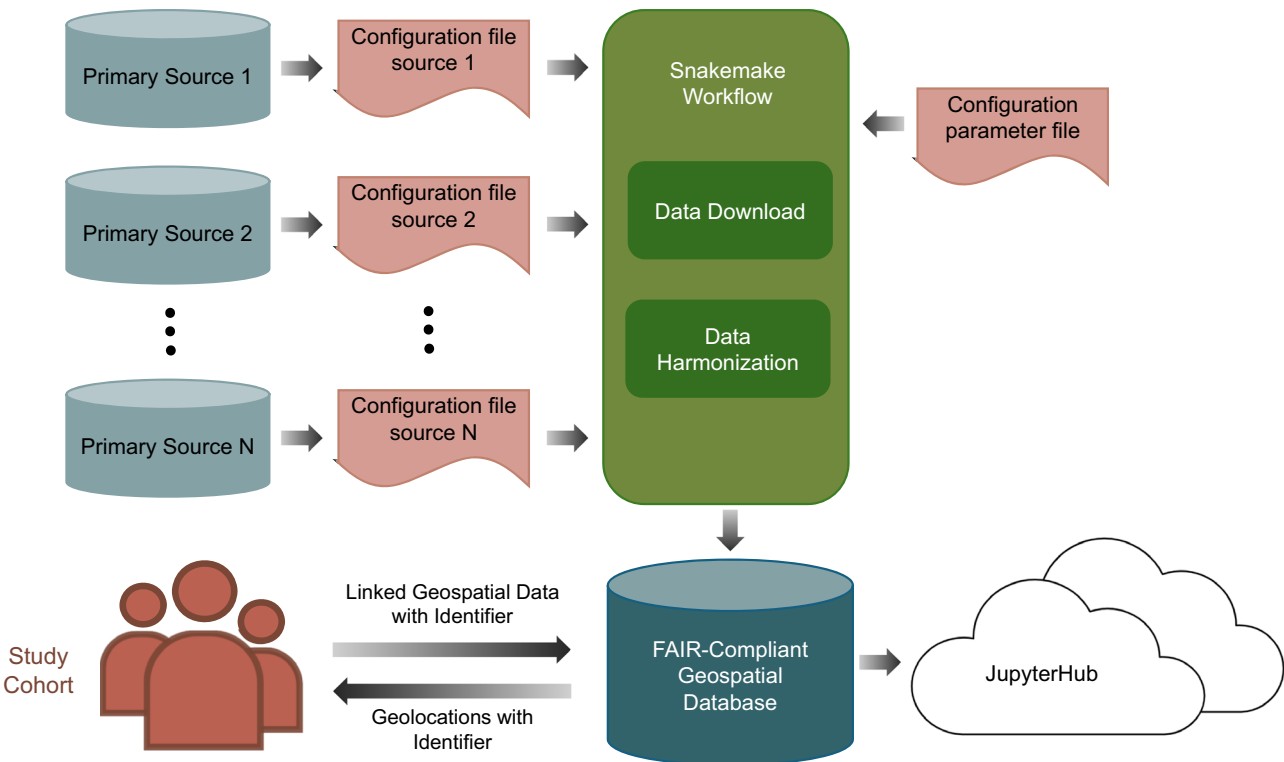

**Fig. 2 | Schema of the CLUES Framework.** This diagram illustrates the workflow of the CLUES framework, starting from multiple primary data sources, each configured by a respective configuration source. The data is then processed through a Snakemake workflow that includes steps for data download, processing, and harmonization. The processed data is stored in a customized geospatial database and made accessible via JupyterHub. Finally, the database can be linked to the study of interest using participant geolocations with identifier. FAIR; Findability, Accessibility, Interoperability, Reusability.

which the harmonised outputs are stored in a structured geospatial database.

In CLUES, data harmonisation ensures that all environmental layers conform to a consistent spatial extent, temporal coverage, and organisational structure. During execution, the source-specific Python download and processing scripts apply the harmonisation settings by retrieving the required spatial and temporal subsets and stitching tiled products into seamless rasters where necessary. Most datasets integrated in CLUES are natively provided in WGS84. MODIS vegetation indices (NDVI, EVI) are an exception, as they are distributed in a sinusoidal equal-area projection. All datasets are retained in their original CRS.

The workflow standardises directory organisation, naming conventions, and output formats (GeoTIFF, netCDF, CSV), ensuring that heterogeneous datasets can be queried consistently throughout the pipeline. During the subsequent linkage step, CLUES transforms participant coordinates, provided in geographic latitude-longitude, into the native CRS of each raster before extracting environmental values, enabling accurate point-level enrichment without resampling large raster datasets. Participant locations are expected to be provided in geographic latitude–longitude (WGS84). If location data are available in a different coordinate reference system, users must reproject them to WGS84 prior to linkage.

## Configuration Management
CLUES uses a configuration system that separates global workflow settings from source-specific parameters. A general workflow configuration file defines spatial and temporal extent, storage locations, and update settings. Each environmental data source is governed by an independent, source-specific configuration file that specifies the datasets to be retrieved, along with associated metadata and optional neighbourhood-level processing (e.g., mean, standard deviation, or

terrain-derived metrics). These processing options are described in detail in the following section.

The advantages of this modular design include fully reproducible data-generation across runs and the ability to extend or refine environmental datasets without altering the core workflow, supporting transparent versioning for downstream analyses. Complete configuration files and parameter descriptions are provided in CLUES online documentation. Further details are provided in the Supplementary Information (SI 2).

## Spatial Context and Environmental Metrics
To facilitate spatial context analyses, the CLUES workflow computes neighbourhood-level metrics at multiple distances (radii) using several data processing methods. These metrics help capture environmental influences at different spatial scales and embed local context into geospatial datasets.

CLUES employs mean and standard deviation filters with circular kernels of different sizes, which effectively reduce noise and create a smoothing effect or identify regions with high texture or contrast, while the variance filter enhances the detection of regions with significant variations in intensity, further improving the accuracy of the analysis. Smaller kernels highlight local variation, while larger ones capture broader context. Currently, these metrics are applied to digital elevation models (DEMs), World Settlement Footprint, and tree cover data. Additionally, the Zevenbergen-Thorne[38] algorithm is utilized on DEMs to derive topographic features such as slope (terrain steepness) and aspect (direction the slope faces).

CLUES applies fixed-radius circular kernels, with kernel sizes defined by the user in the configuration file. Adaptive bandwidth kernels[39] are not part of the core workflow, as CLUES operates on continuous raster surfaces rather than irregular point patterns, where adaptive approaches typically offer the greatest benefit. However, an

optional stand-alone script is provided in the online documentation for users who wish to apply adaptive smoothing to external datasets.

## Data formats and software ecosystem

Within the framework, the appropriate file format is essential for efficient data storage, retrieval, and processing. The workflow provides geospatial data in three distinct formats, depending on the nature of the geospatial asset. Firstly, netCDF (Network Common Data Form) is a widely used format for array-oriented scientific data. It supports the creation, access, and sharing of data in a self-describing, portable, and scalable manner. NetCDF is particularly popular in the climate and meteorological communities due to its ability to handle large datasets and its compatibility with various software tools. Secondly, GeoTIFF is a widely used file format for geospatial raster data. It extends the TIFF (Tagged Image File Format) by including geographic metadata, allowing for precise georeferencing of raster images. Lastly, CSV (Comma-Separated Values) files are a simple yet powerful format for tabular data, used to provide information on socioeconomic features. Additional information on data formats, together with utility scripts to optionally convert socioeconomic CSV indicators into GIS-compatible formats (e.g., GeoPackage), is available in the online documentation (https://bih-dmbs.github.io/CLUES/data_formats/).

The workflow employs Python, one of the dominant programming languages in scientific community, due to its vast ecosystem of libraries. Snakemake is a workflow management system that simplifies the execution of complex data analysis pipelines. It uses a human-readable syntax to define workflows, ensuring reproducibility and scalability. Snakemake is particularly useful for managing dependencies and parallelizing tasks, making it an invaluable tool for large-scale data processing. GeoPandas extends the capabilities of the popular Pandas library to handle geospatial data. It provides data structures and operations for manipulating geographic information. GeoPandas supports various file formats and offers functionalities for geometric operations as spatial joins and overlays. Rasterio is a library designed for reading and writing geospatial raster data. It provides a simple and efficient interface for accessing raster files, such as GeoTIFFs. Rasterio supports various raster operations, such as resampling, cropping, and masking, making it an essential tool for geospatial analysis. Rioxarray and xarray are libraries that facilitate the manipulation of multi-dimensional arrays. Xarray provides data structures for labelled arrays, enabling the handling of complex datasets with ease. Rioxarray extends xarray's capabilities by adding support for raster data, allowing for seamless integration with other geospatial libraries. The netCDF4 library provides an interface for reading and writing netCDF files. Non-direct data retrieval relies on OWSLib a Python package designed for OGC web service interface standards, including WMS or CDS API client library for accessing geospatial data.

## Prerequisites for implementation

The CLUES framework has been designed to be accessible to the broad scientific community, including users with limited technical expertise. Its primary goal is to facilitate the integration of geospatial data into research for scientists eager to leverage such data. The protocol requires only a basic understanding of the Python programming language and access to a computer with a stable internet connection. Comprehensive code, example datasets, and detailed documentation are provided, eliminating the need for users to write code from scratch. For those wishing to extend the framework beyond the provided code, familiarity with the Linux or Windows command line, bash scripting, setting up virtual environments, and submitting jobs to high-performance computing (HPC) clusters would be advantageous.

The required computing infrastructure includes a Linux or Windows machine or a HPC system, with sufficient storage capacity to accommodate the downloaded geospatial database.

## Step-by-step guide: from setup to execution

Deploying CLUES in a production environment involves several key steps. First, the repository must be cloned and a dedicated Python virtual environment initialized with all required dependencies. After obtaining third-party credentials (e.g., API keys) and configuring the necessary accounts, users can tailor the configuration files to suit their specific research objectives. The Snakemake workflow is then executed to build the geospatial database. Once the database is generated, health data can be securely linked using the provided integration scripts. All necessary steps are detailed in the following sections and are also documented in the github page of CLUES framework (https://bih-dmbs.github.io/CLUES/).

To set up the environment, clone the GitHub repository using the following commands. This repository contains the necessary code, configuration files and example data. Then install the Python packages using pip and import them into the Python environment, as follows:

```
# Navigate to the target working directory
# Clone the GitHub repository
git clone https://github.com/BIH-DMBS/CLUES.git
# Set up a virtual python environment
python -m venv CLUESEnv
# Activate created virtual python environment
CLUESEnv\Scripts\activate
# Install the Python packages using pip
pip install -r workflows\requirements.txt
```

To configure the CLUES framework the general workflow configuration file (config\config.json) needs to be customized. The "key":-value pairs in the file should be self-explanatory and are described in detail in the CLUES documentation (https://bih-dmbs.github.io/CLUES/) as well. Key parameters include:

- "download_folder": the path to the folder where the CLUES database will be stored.
- "tmp_folder": the path to the folder for temporary files that will be created during the database generation.
- "configs_assets_folder": the path to the folder containing configuration files for the primary data sources. These files are explained later in the document.
- "secrets_folder": the path to the folder where credentials are stored. Details on credential generation and handling are provided in a later section.
- "years" and "update_years": list of years specifying either the years for which the data should be downloaded or, in case a database already exists, the years that should be updated based on data presently available from the primary sources.

Example of the config.json file

```
{
    "download_folder": "\\CLUES_database",
    "tmp_folder": "CLUES\\tmp",
    "configs_assets_folder": "CLUES\\CLUES_utils\\configs_sources",
    "secrets_folder": "CLUES\\CLUES_utils\\secrets",
    "years": ["2022","2023","2024","2025"],
    "update_years": ["2025"],
    "area": "Germany",

}
```

- "area": specifies the key corresponding to a predefined bounding box in the bbox.json file (a sample entry is shown below), located in the same folder. A bounding box is a rectangular area defined by four coordinates: the minimum longitude, minimum latitude, maximum longitude, and maximum latitude. These coordinates

represent the geographical boundaries of a region. For example, the bounding box for Europe is represented as "Europe": [72, −15, 30, 42.5]. Here, 72 is the minimum longitude, −15 is the minimum latitude, 30 is the maximum longitude, and 42.5 is the maximum latitude. This structure helps in identifying and visualizing the geographical extent of Europe on a map. The bbox.json file contains a set of predefined bounding box for several areas. If a bounding box is not available for a specific area, it can be obtained using online tools such as bboxfinder.com or OpenStreetMap.

It is important to note that certain features within CLUES are only available for specific years. If the specified "key":value pair "years" does not include any of these years, no data will be downloaded. Similarly, certain datasets are regional restricted. For example, Corine Land Cover data is only available for Europe; if regions outside Europe are requested, the data will not be downloaded.

bbox.json

```
{
"Europe": [72, −15, 30, 42.5],
"UK": [60.8608, −8.6493, 49.9096, 1.7689],
"Brandenburg": [53.5587, 11.2682, 51.3618, 14.7636],
…
"Berlin": [52.7136,12.9673,52.2839,13.816]
}
```

Some of the primary data sources may require the user to have an account to get data access. This pertains access to the CDS API Copernicus and NASA. For data associated to Copernicus a European Centre for Medium-Range Weather Forecasts (ECMWF) account is necessary to create a personal access token for access to atmosphere data store and climate data. Go to ECMWF | Advancing global NWP through international collaboration and create account. For both services a CDS API personized access token must be generated. To do so, visit CDSAPI setup - Climate Data Store and CDSAPI setup - Atmosphere Data Store. To access data associated with NASA earthdata (https://www.earthdata.nasa.gov/) credentials are required. The login keys/tokens must be stored at the location defined in the config.json file. The tokens must be saved in the following files: cdsapirc_climate.sct, cdsapirc_atmo.sct, and nasa.sct, as shown below. These files need to be put into the "secrets" folder defined in the config.json file.

cdsapirc_climate.sct
```
url: https://cds.climate.copernicus.eu/api
key: b7abef0f-xxxx-xxxx-xxxx-924a73789ad3
```

cdsapirc_atmo.sct
```
url: https://ads.atmosphere.copernicus.eu/api
key: b7abef0f-xxxx-xxxx-xxxx-924a73789ad3
```

nasa.sct
```
token: eyJ0eXAiO…_RqCIofBuA
```

Each primary data source has a dedicated JSON configuration file that specifies the data to be downloaded. These configuration files, available in the git repository, allow for customization of the downloaded data. By editing these files and removing certain items, users can tailor the database to their needs. Each configuration file is unique due to the differences in the primary data sources. If the files remain unmodified, they provide access to all the environmental features listed in the documentation (https://bih-dmbs.github.io/CLUES/dataCatalog/) of the CLUES database. The files are located in the folder workflows\config_sources\. The source-specific configuration files contain metadata about the primary sources, including the URL, temporal resolution, type, and a list of available geospatial attributes used by CLUES. These attributes are connected to the key 'variables'

from the primary source. For example, in the JSON configuration file that manages the download of CAMS global reanalysis (EAC4) data from Copernicus, the listing of the assets of interest as atmospheric $SO_2$ or $O_3$ levels is associated with this key.

To initiate the workflow for CLUES, users need to start by setting up their environment and ensuring all necessary dependencies are installed. The workflow is managed using Snakemake. Users can start the workflow by navigating to the directory containing the workflow files. From the command prompt, the workflow can be started using: snakemake -s.\workflows\snakefile --cores 16 -p. This command specifies the use of the Snakemake file located in the \workflows directory, allocates cores for parallel processing, and enables the printing of detailed progress information.

```
# Activate virtual environment
CLUESEnv\Scripts\activatevs
# Start workflow
snakemake -s.\workflows\snakefile --cores 16 -p
```

The workflow may occasionally terminate with an error, such as when one of the primary sources is down, for example due to scheduled maintenance. In such cases, the workflow needs to be restarted using the command: snakemake -s.\workflows\snakefile --cores 16 -p --rerun-incomplete. The --rerun-incomplete flag ensures that any incomplete steps from the previous run are attempted again. For more detailed information about the cause of the error, corresponding log files are available. Occasionally, delays may occur if primary data sources are temporarily unavailable. Such interruptions are expected, as the workflow depends on continuous access to these external sources. If a file is found to be corrupted while working with the database, it can be simply deleted; restarting the workflow will automatically download the file anew.

In addition to the core workflow that generates the database, the framework includes scripts to enrich location data and auxiliary Jupyter notebooks that serve as tutorials for using and interacting with the database.

```
python link_locations.py locations.csv input_folder out
put_folder
```

The link_locations.py script is designed to process netCDF and GeoTIFF files located in the input folder by linking the locations specified in the location.csv file to the corresponding geospatial features. The location.csv file should contain at least three columns: longitude, latitude, and subject ID. The results of this enrichment process are saved in the output folder, with JSON files generated for features derived from netCDF, and CSV files for those derived from GeoTIFFs. The output for netCDF-derived files is optimised for minimizing file size while maintaining essential temporal and spatial information. Since netCDF files often contain time-series data with relatively coarse spatial resolution, subjects linked to the same pixel are grouped together, and the corresponding time-series data is stored only once. In addition, a set of Jupyter notebooks is included in the CLUES framework and is available at the CLUES-Git (https://github.com/BIH-DMBS/CLUES/tree/main/notebooks). The notebooks give examples on how the downloaded data can be used, visualized and linked to subject locations. The different datatypes and formats are covered in these demonstrators. Similarly, notebooks are available that cover all types of data included in the CLUES, ensuring comprehensive guidance for users.

Beyond the default data catalogue, additional datasets can be integrated as new environmental products, satellite missions, or thematic data layers become available, or as project-specific research questions require different exposures. CLUES is actively extended with new sources, and users may incorporate datasets of their choice by

following the dedicated tutorial in the online User Guide. The tutorial demonstrates how to create a source-specific configuration file, register the new source within the Snakemake workflow, and implement the corresponding download function.

All datasets integrated into CLUES are open-source and publicly available. Each primary data source is subject to its own terms of use. Source-specific license information and reference links are listed in the online Data Catalogue. Users should review these usage policies prior to data use to ensure correct attribution and compliance.

### Reporting summary

Further information on research design is available in the Nature Portfolio Reporting Summary linked to this article.

## Data availability

No new datasets were generated during this study. CLUES retrieves publicly available environmental datasets from third-party providers, and all data sources used in this workflow are openly accessible. Where available, DOIs are provided for each dataset in the CLUES documentation, along with access links and licensing information for each data source.

## Code availability

All code is available in Python format on GitHub (https://github.com/BIH-DMBS/CLUES/) (DOI: [https://doi.org/10.5281/zenodo.19092838], 2026).

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

## Acknowledgements

This work contributes to the objectives of the Earth Brain Health Commission, an interdisciplinary initiative addressing the impacts of environmental megatrends, including climate change, urbanicity, pollution and social disparity, on brain and mental health. Funded by the European Union (Grant agreement No 101057429). Complementary funding was received by UK Research and Innovation (UKRI) under the UK government's Horizon Europe funding guarantee (10131373 and 10038599) and the National Key R&D Program of Ministry of Science and Technology of China (MOST 2023YFE0199700). Views and opinions expressed are however those of the author(s) only and do not necessarily reflect those of the European Union, the European Health and Digital Executive Agency (HADEA), UKRI or MOST. AM gratefully acknowledges support from the European Research Council under a Consolidator grant (101001118). Neither the European Union nor HADEA nor UKRI nor MOST can be held responsible for them.

## Author contributions

Conceptualization, MJ, ST, GS, AM, EP; Investigation and Data curation, MJ, EP, PR, SH, KS, and GS; Methodology, MJ and ST; Software, MJ, ST, PR, and KS; Writing – original draft, MJ, ST, EP, and PR; Writing – review & editing, MJ, ST, EP, PR, KS, SH, GS, AM, and RE; Supervision, ST.

## Funding

## Competing interests

The authors declare no competing interests.

## Additional information

## the environMENTAL consortium

Rieke Aden[10], Kofoworola Agunbiade[11], Ole A. Andreassen[12], Helga Ask[13,14], Anastasios-Polykarpos Athanasiadis[15], Tobias Banaschewski[16,17], Antoine Bernas[5], Sarah J. Böttger[18], Ragnhild Brandlistuen[19], Vince D. Calhoun[20], Xiao Chang[21], Di Chen[11], Nina Christmann[16], Isabelle Claus[22], Nicholas Clinton[23], Yuxiang Dai[21], Sylvane Desrivières[11], Francisco José Eiroa-Orosa[24], Guillem Feixas[24], Sara Fernández-Cabello[12,13], Andreas J. Forstner[22], Jaime Gallego[24], Stefanie Heilmann-Heimbach[22,25], Andreas Heinz[6,26,27], Esther Hitchen[7], Per Hoffmann[22,25], Nathalie Holz[16], Reiya Itatani[24], Karina Janson[10,16], Tianye Jia[21], Viktor Jirsa[15], Hedi Kebir[7], Rikka Kjelkenes[12,13], Vanessa Köhler[28], Tristram A. Lett[7], Yuzhu Li[21], Carina M. Mathey[22], Andreas Meyer-Lindenberg[17,29], Abigail J. Miller[22,25], Frauke Nees[30], Maja Neidhart[10,16,31], Gaia Novarino[32], Markus M. Nöthen[22], George Ogoh[33], Myrto Patraskaki[34], Charlie Pearmund[35], Spase Petkoski[15], Markus Ralser[36,37,38], Michael A. Rapp[18], Jean-Charles Roy[7], Tamara Schikowski[39], Karen Schmitt[34], Ameli Schwalber[28], Emanuel Schwarz[17,29,40], Tatjana Schütz[1], Beke Seefried[10], Emin Serin[31,41,42], Mel Slater[24], Peter Sommer[34], Bernhard Spanlang[35], Bernd C. Stahl[33], Ulrike H. Taron[1], Paul Thompson[43], Heike Tost[17,29], Mira Tschorn[18], Nilakshi Vaidya[7],

Dennis van der Meer[12], Henrik Walter[6,31], Lars T. Westlye[12,13], Johannes H. Wilbertz[34], Yunman Xia[21], Allan H. Young[44], Xinyang Yu[11], Jiacan Yuan[45], Yanqing Zhang[46] & Zuo Zhang[11,47]

[10]Institute of Medical Psychology and Medical Sociology, University Medical Center Schleswig-Holstein, Kiel, Germany. [11]Social, Genetic and Developmental Psychiatry Centre, King's College London, London, United Kingdom. [12]Centre for Precision Psychiatry, Division of Mental Health and Addiction, Oslo University Hospital & Institute of Clinical Medicine, University of Oslo, Oslo, Norway. [13]Department of Psychology, University of Oslo, Oslo, Norway. [14]PsychGen Centre for Genetic Epidemiology and Mental Health, Norwegian Institute of Public Health, Oslo, Norway. [15]Institut de Neurosciences des Systèmes, Aix-Marseille Université, Marseille, France. [16]Department of Child and Adolescent Psychiatry and Psychotherapy, Central Institute of Mental Health, Medical Faculty Mannheim Heidelberg University, Mannheim, Germany. [17]German Center for Mental Health (DZPG), Mannheim-Heidelberg-Ulm site, Mannheim, Germany. [18]Social and Preventive Medicine, University of Potsdam, Potsdam, Germany. [19]Department of Child Health and Development, Norwegian Institute of Public Health, Oslo, Norway. [20]Tri-institutional Center for Translational Research in Neuroimaging and Data Science (TReNDS), Georgia State University, Georgia Institute of Technology, and Emory University, Atlanta, GA, USA. [21]Institute of Science and Technology for Brain-Inspired Intelligence, Fudan University, Shanghai, China. [22]Institute of Human Genetics, University of Bonn School of Medicine and University Hospital Bonn, Bonn, Germany. [23]Google LLC, Mountain View, CA, USA. [24]Departament de Psicologia Clínica i Psicobiologia, Event Lab, Institute of Neuroscience, University of Barcelona, Barcelona, Spain. [25]Life and Brain GmbH, Bonn, Germany. [26]Department of Psychiatry and Psychotherapy, University of Tübingen, Tübingen, Germany. [27]German Center for Mental Health (DZPG), Tübingen site, Tübingen, Germany. [28]concentris research management GmbH, Fürstenfeldbruck, Germany. [29]Department of Psychiatry and Psychotherapy, Central Institute of Mental Health, Medical Faculty Mannheim Heidelberg University, Mannheim, Germany. [30]Institute of Medical Psychology, Ludwig-Maximilians-Universität München, Munich, Germany. [31]Department of Psychiatry and Psychotherapy, Charité – Universitätsmedizin Berlin, Berlin, Germany. [32]Institute of Science and Technology Austria (ISTA), Klosterneuburg, Austria. [33]School of Computer Science, University of Nottingham, Nottingham, United Kingdom. [34]Ksilink, Strasbourg, France. [35]Virtual Bodyworks, Barcelona, Spain. [36]Department of Biochemistry, Charité – Universitätsmedizin Berlin, Berlin, Germany. [37]Nuffield Department of Medicine, University of Oxford, Oxford, United Kingdom. [38]Max Planck Institute for Molecular Genetics, Berlin, Germany. [39]IUF Leibniz Research Institute for Environmental Medicine, Düsseldorf, Germany. [40]Hector Institute for Artificial Intelligence in Psychiatry, Central Institute of Mental Health, Medical Faculty Mannheim, Heidelberg University, Mannheim, Germany. [41]Einstein Center for Neurosciences Berlin, Charité – Universitätsmedizin Berlin, Berlin, Germany. [42]Bernstein Center for Computational Neuroscience Berlin, Berlin, Germany. [43]Stevens Neuroimaging and Informatics Institute, Keck School of Medicine, University of Southern California, Los Angeles, CA, USA. [44]Department of Psychological Medicine, King's College London and South London and Maudsley NHS Foundation Trust, London, United Kingdom. [45]Department of Atmospheric and Oceanic Sciences, Institute of Atmospheric Sciences, Fudan University, Shanghai, China. [46]Institute of Biomedical Sciences (IBS), Fudan University, Shanghai, China. [47]Institute for Mental Health, School of Psychology, University of Birmingham, Birmingham, United Kingdom.

