## [Transparent Peer Review file · Nature Communications]

CLUES A Comprehensive Workflow for Integrating Geospatial Data in Biomedical Research

Corresponding Author: Dr Sven Twardziok

Version 0:

Reviewer comments:

Reviewer #1

(Remarks to the Author)

The manuscript CLUES: A Comprehensive Workflow for Integrating geospatial Data in Biomedical Research describes a computational system designed to acquire and consolidate geospatial information from open sources (e.g., mainly satellite data and related products) in order to allow efficient matching with biomedical data comprising location, spatial profiles and movement profiles.

In general, the topic is interesting, however, there is little novelty to this manuscript beyond the integration of various open source components in order to create a suitable system. The venue for impact might be that - if the system is easy enough to adopt and adapt - it can be a first step towards FAIRness for geospatial data fusion with datasets from different domains.

The paper goes very far into details that are commonly not communicated through research papers, for example, details of software configuration and deployment.

What the paper is actually missing is a scientific question and a related evaluation. The system proposes quite some innovative aspects, but a formal evaluation on

a) performance

or

b) user adoption cost (e.g., doing a user test with fellow researchers that shall integrate some datasets with some spatial data)

Thus, I first recommend to remove all the configuration details with a paragraph linking to a tutorial hosted alongside the source code on github.

Second, I recommend to either put scalability or usage complexity at the heart of the proposal and present either a scalability test with multiple use cases or a medium-scale user test.

In the current form, I would not recommend this paper for publication.

In summary: the system is interesting, the paper just does not present any research or evaluation regarding the system.

(Remarks on code availability)

Reviewer #2

(Remarks to the Author)

This article describes a digital workflow - CLUES - that facilitates the selection, download, and preparation of geospatial

datasets with the goal of facilitating medical and public health studies aiming to quantify environmental exposures. There is a clear need for this type of framework, and CLUES has been developed using the FAIR principles, and with individual-level data in mind, ensuring a local deployment of the tool that can be fully secured at the user's end. The manuscript is well written, with a good background introduction and justifications for the framework. CLUES has the potential to be used in numerous applications, and not constrained to the type of research (mental health) for which it has been initially developed.

I do not have major asks for edits, but I'm listing below several comments and minor edits that could benefit a revision of the article:

- It was not clear to me how an additional data set could be incorporated in the existing framework. For example, if I would like to add the openly available WorldPop population distribution raster (very much used in socio-economic health studies) within CLUES, what would this entail? What level of coding would be needed? I encourage you to give such an example, or at least to briefly explain how it would be done and point to a section in your technical documentation that gives such example (with code)

- There is no clear mention of how the database that CLUES generates can be linked to the pieces of software or modeling framework that will be used to analyse exposure. I know this is not the aim of the article, but it would be nevertheless very useful for the reader to understand how this step is eased by CLUES and what statistical tools have been used in your previous studies using CLUES.

- You are stating in your introduction that you are standardizing projections. But there is no mention about how you do it. Are you reprojecting datasets? This is easy for vector data sets, but as you know more tricky for raster datasets. So how are you doing it? Can the user select the projection to be used?

I understand that for exposome type of studies, you just need to get the exposition variable values under the location of the individual data, so projection would not matter too much in this case. However, because you are stating that CLUES can be used in many other settings, how data are projected could matter greatly. If the projected data sets are used for zonal statistics over large areas, or for geographical accessibility modeling, having an equal area metric projection would be very important, for example. I would like to see some more details on this.

My comments actually relate broadly to the box "Data harmonization" in Fig 3, because it's not very explicit how exactly datasets are harmonized.

- I gather that you are not resampling raster data set within the workflow, but this is not really explicit and this could be mentioned somewhere.

- Line 145-146: how does the CLUES facilitate the linkage with geolocation, especially when you have residential address on one side (individual), and coordinates on the other (exposure variables)? Also, you mention "routes". What do you mean exactly? Tracked individual's data along which you want the exposure values? Specify

- In the section "Usage policies", can you specify whether the "terms of use" of each data set are downloaded as well and readily available to the user. If they are not, that would be very cumbersome for the user to get them individually ! Also, it would be great if an AI tool could correctly summarize all terms of use of downloaded data sets to flag anything of interest for the user.

(Remarks on code availability)

I have not reviewed the code source itself, but I have reviewed the github repository and it seems complete with user guide, license, config files, etc.

Reviewer #3

(Remarks to the Author)

This manuscript proposed a framework to integrate geospatial datasets in biomedical research settings. They claim it is the first open-source tool to generate individual-level exposure data, while it's not necessary to have any experience working with geospatial data. I believe that the proposed framework is a nice idea and could be very beneficial to public health research, but some concerns need to be addressed first.

1. CLUES currently requires users to have a working knowledge of Python and programming fundamentals. Have you considered developing an R implementation, given R's popularity in spatial statistics and its relatively user-friendly interface for statistical users

1a. Or alternatively, a graphical user interface (GUI) to enhance accessibility for users with limited programming experience?

2. The description suggests that CLUES applies mean and standard deviation filters using circular kernels of varying sizes to capture local and broader context. Could you clarify whether these kernels represent multiple fixed spatial scales or if an adaptive (variable-width) bandwidth approach is used? If fixed, have you considered implementing adaptive kernels to better account for spatial heterogeneity in data density?

3. You mention that socioeconomic data are provided as CSV files. Could you elaborate on the decision to use CSV rather than a geospatially compatible tabular format such as DBF or GeoPackage? CSVs can lack important metadata, field type definitions, and direct compatibility with shapefiles, which may introduce issues with field name truncation or type conversion during import. A DBF or similar format could improve interoperability with common GIS software and preserve data integrity.

4. Could the authors clarify how easily CLUES can be adapted to other regions, such as the U.S. or low- and middle-income

countries, where equivalent data infrastructures and APIs may differ or be unavailable? What level of customization would be required for non-European applications?

5. Given that CLUES harmonizes coordinate reference systems and bounding boxes across diverse sources, how does the workflow handle discrepancies in datum definitions (e.g., WGS84 vs. ETRS89) and projection distortions at global scales? Has this been tested outside of Europe?

6. The privacy-preserving linkage approach is commendable. However, could the authors elaborate on how CLUES ensures compliance with non-EU data protection frameworks (e.g., HIPAA in the U.S.)? Are there built-in options for differential privacy or coordinate perturbation beyond rounding or masking?

(Remarks on code availability)

The github and code provided is very reproducible for those who have programming experience in Python. The tutorials are also great to get started and follow along with some of their examples in Europe. All important documentation are included, such as README files, instructions, and comments throughout scripts.

Version 1:

Reviewer comments:

Reviewer #2

(Remarks to the Author)

Thank you for having addressed all of my comments very well. I particularly appreciated the efforts of adding several new sections and examples in the user guide, which will facilitate adoption of users.

I have now further comments.

(Remarks on code availability)

Reviewer #3

(Remarks to the Author)

The authors did a nice job addressing my comments and I am satisfied with their responses and revisions.

I do agree with Reviewer #1 that the paper does not present any research or evaluation regarding the system. The paper should be published, in my opinion, but maybe Scientific Data (Nature) is a better fit. Nature Communications is more research-oriented. Will leave it up to the editors.

(Remarks on code availability)

Code was already well documented and reproducible after the first round of review. Revised remarks have been added for clarity and the workflow reads even better.

We would like to thank the Editorial team and all Reviewers for their insightful comments and time. We revised the manuscript in line with the reviewer's comments, followed by our point-by-point responses, presented in blue. In addition, we have revised it according to the editorial guidelines.

Reviewer #1 (Remarks to the Author):

The manuscript CLUES: A Comprehensive Workflow for Integrating geospatial Data in Biomedical Research describes a computational system designed to acquire and consolidate geospatial information from open sources (e.g., mainly satellite data and related products) in order to allow efficient matching with biomedical data comprising location, spatial profiles and movement profiles.

In general, the topic is interesting, however, there is little novelty to this manuscript beyond the integration of various open source components in order to create a suitable system. The venue for impact might be that - if the system is easy enough to adopt and adapt - it can be a first step towards FAIRness for geospatial data fusion with datasets from different domains.

The paper goes very far into details that are commonly not communicated through research papers, for example, details of software configuration and deployment.

What the paper is actually missing is a scientific question and a related evaluation. The system proposes quite some innovative aspects, but a formal evaluation on

a) performance

Add one sentence with current research activities into the discussion section of the main document

or

b) user adoption cost (e.g., doing a user test with fellow researchers that shall integrate some datasets with some spatial data)

1. Thus, I first recommend to remove all the configuration details with a paragraph linking to a tutorial hosted alongside the source code on github.

We agree that the configuration details may be too detailed for the purposes of the manuscript and have updated the text accordingly (lines 449–460):

“CLUES uses a configuration system that separates global workflow settings from source-specific parameters. A general workflow configuration file defines spatial and temporal extent, storage locations, and update settings. Each environmental data source is governed by an independent, source-specific configuration file that specifies the datasets to be retrieved, along with associated metadata and optional

neighbourhood-level processing (e.g., mean, standard deviation, or terrain-derived metrics). These processing options are described in detail in the following section.

The advantages of this modular design include fully reproducible data-generation across runs and the ability to extend or refine environmental datasets without altering the core workflow, supporting transparent versioning for downstream analyses. Complete configuration files and parameter descriptions are provided in CLUES online documentation. Further details are provided in the Supplementary Information (SI 2)”

2. Second, I recommend to either put scalability or usage complexity at the heart of the proposal and present either a scalability test with multiple use cases or a medium-scale user test.

We thank the reviewer for this suggestion. We have expanded the evaluation of CLUES to include a systematic and quantitative assessment of scalability and performance.

Specifically, we assessed:

1. Scalability of environmental database generation across geographic extents
1. We benchmarked CLUES performance using three countries with different spatial extent, such as France (1,247,000 km²), Nepal (362,000 km²), and Suriname (210,000 km²), to demonstrate scalability across diverse geographic sizes and global contexts.
2. Temporal scalability and incremental database expansion
For each region, we generated environmental datasets starting with a single year and progressively added additional years. At each step, we recorded total processing time and storage usage, distinguishing between the initial setup phase and subsequent incremental updates.
3. Linking performance for GeoTIFF-based raster data
We assessed the performance of point-based linkage to GeoTIFF rasters using synthetically generated datasets with varying spatial resolutions and increasing numbers of point locations. Execution time was measured as a function of raster size and sample size.
4. Linking performance for netCDF-based raster data
We evaluated linkage performance for netCDF files using a CLUES-generated environmental dataset covering the Scandinavian peninsula (1,831,726 km²) over the period 2000–2025. Linking runtimes were recorded for 1,000 and 10,000 locations and analysed in relation to file size.

We have incorporated the results of these evaluations in the Supplementary Information (SI1 Scalability and Performance evaluation), as indicated by the editor, and refer the reader to this material in the current section “Performance, Reproducibility, and Interoperability” (lines 204–206):

“Additional performance and scalability benchmarks, including multi-country runtime comparisons and controlled raster-linkage evaluations, are provided in the Supplementary Information (SI 1).”

In addition, we have revised the Introduction (lines 128–131) to clearly state the aims of CLUES and to introduce the evaluation of its scalability and performance”

“In this work, we demonstrate the application of CLUES and describe the structure and expected outputs of the framework, followed by an evaluation of its performance. CLUES delivers reproducible, scalable, and efficient performance across diverse environmental datasets, resolutions, and cohort settings.”

In the current form, I would not recommend this paper for publication.

In summary: the system is interesting, the paper just does not present any research or evaluation regarding the system.

Reviewer #2 (Remarks to the Author):

This article describes a digital workflow - CLUES - that facilitates the selection, download, and preparation of geospatial datasets with the goal of facilitating medical and public health studies aiming to quantify environmental exposures. There is a clear need for this type of framework, and CLUES has been developed using the FAIR principles, and with individual-level data in mind, ensuring a local deployment of the tool that can be fully secured at the user's end. The manuscript is well written, with a good background introduction and justifications for the framework. CLUES has the potential to be used in numerous applications, and not constrained to the type of research (mental health) for which it has been initially developed. I do not have major asks for edits, but I'm listing below several comments and minor edits that could benefit a revision of the article:

1. It was not clear to me how an additional data set could be incorporated in the existing framework. For example, if I would like to add the openly available WorldPop population distribution raster (very much used in socio-economic health studies) within CLUES, what would this entail? What level of coding would be needed? I encourage you to give such an example, or at least to briefly explain how it would be done and point to a section in your technical documentation that gives such example (with code)

We thank the reviewer for highlighting this important point. We would like to note that we are actively expanding the default CLUES data catalogue, as new or higher-quality environmental products become available. For example, we are currently working on integrating higher resolution datasets, such as the CAMS European atmospheric composition data (DOI: 10.24381/7cc0465a) and the GlobalHighAirPollutants (GHAP)

datasets (DOIs: 10.5281/zenodo.10800980, 10.5281/zenodo.10208188, 10.5281/zenodo.7397059, 10.5281/zenodo.14207363),

We have also already integrated the higher resolution ERA5 climate datasets (DOI: 10.24381/cds.e2161bac), and the WorldPop population distribution dataset, motivated by the reviewer's suggestion.

We have added a dedicated tutorial to the online CLUES documentation (<https://bih-dmbs.github.io/CLUES/UserGuide/#adding-a-new-dataset-to-clues-worldpop-example>). The tutorial provides a step-by-step example based on the WorldPop dataset and demonstrates:

- How to inspect available products (via the WorldPop API),
- How to create a source-specific configuration file,
- How to register the new dataset in the Snakemake workflow, and
- How to implement a small download script.

The documentation also specifies the level of coding required. We clarify that running the default CLUES workflow, including the new datasets that we prespecified, requires only minimal Python familiarity. Expanding CLUES with new datasets may require basic to intermediate Python familiarity, depending on the source, but the tutorial and code examples are intended to make this process accessible to users.

For better deployment, we also added a working, containerized Docker script to ensure consistent environments across development and production.

We have added a new subsection titled "Extensibility and new data sources" to the manuscript (lines: 722–729):

"Extensibility and new data sources

Beyond the default data catalogue, additional datasets can be integrated as new environmental products, satellite missions, or thematic data layers become available, or as project-specific research questions require different exposures. CLUES is actively extended with new sources, and users may incorporate datasets of their choice by following the dedicated tutorial in the online User Guide. The tutorial demonstrates how to create a source-specific configuration file, register the new source within the Snakemake workflow, and implement the corresponding download function."

2. There is no clear mention of how the database that CLUES generates can be linked to the pieces of software or modeling framework that will be used to analyse exposure. I know this is not the aim of the article, but it would be nevertheless very useful for the reader to understand how this step is eased by CLUES and what statistical tools have been used in your previous studies using CLUES.

We agree that adding information on how the CLUES-derived data can be incorporated into downstream modelling frameworks strengthens the manuscript. We have added the following paragraph to the manuscript (lines: 225–238):

“CLUES focuses on the upstream integration and linkage of environmental datasets, while leaving downstream statistical analysis entirely to the user. Environmental layers (e.g., GeoTIFF and netCDF files) can be used directly in standard GIS software (such as QGIS or ArcGIS) or in common analytical environments such as Python (e.g., xarray, rioxarray, rasterio, geopandas) and R (e.g., terra, sf, raster). The linkage step produces flat, analysis-ready exposure tables (CSV and JSON) that can be directly merged with user datasets and imported into any statistical software, including Python, R, or Stata. The linked exposure tables can be analysed using a wide range of methods, including area- or cluster-robust regression models, mixed-effects models, latent-variable models, or machine-learning approaches. To support both workflows, the online documentation includes example notebooks in Python. One set is demonstrating how to load, visualise, and explore generated geospatial layers, and another is illustrating how linked exposure tables can be analysed using a random-forest model on a synthetic dataset.”

3. You are stating in your introduction that you are standardizing projections. But there is no mention about how you do it. Are you reprojecting datasets? This is easy for vector data sets, but as you know more tricky for raster datasets. So how are you doing it? Can the user select the projection to be used?

We agree that our description on projection handling was too brief and we thank the reviewer for raising this point.

CLUES does not globally reproject all datasets to a single target CRS. Most environmental datasets integrated into CLUES are natively provided in WGS84 and are therefore downloaded and stored in their original projection without modification. MODIS vegetation indices (NDVI, EVI) are an exception, as they are only available in the global sinusoidal equal-area projection. For these data, CLUES also keeps the native projection and does not resample them to WGS84 within the workflow.

During the linkage step, CLUES reads the CRS of each raster and reprojects participant coordinates from geographic latitude–longitude (WGS84) into the raster’s native CRS before querying cell values.

Users cannot currently define a single global target projection for all geospatial datasets within CLUES. Therefore, if location data are provided in a projection other than geographic latitude–longitude, users must reproject those locations to WGS84 prior to linkage. In addition, for users who wish to visually overlay multiple raster layers with different projections, reprojection to a common CRS remains necessary. General tutorials for reprojection of raster layers and vector data are available online: <https://py.geocompx.org/06-reproj>.

We have made these clarifications in the manuscript in the Methods (lines 435–447): *“Most datasets integrated in CLUES are natively provided in WGS84. MODIS vegetation indices (NDVI, EVI) are an exception, as they are distributed in a sinusoidal equal-area projection. All datasets are retained in their original CRS.*

... During the subsequent linkage step, CLUES transforms participant coordinates, provided in geographic latitude-longitude, into the native CRS of each raster before extracting environmental values, enabling accurate point-level enrichment without resampling large raster datasets. Participant locations are expected to be provided in geographic latitude–longitude (WGS84). If location data are available in a different coordinate reference system, users must reproject them to WGS84 prior to linkage.”

And Discussion (lines 317–324):

“Finally, for applications that require visual overlay or joint spatial analysis of raster layers from different projections (e.g., WGS84 and MODIS sinusoidal), users may need to reproject datasets to a common CRS. CLUES does not perform this reprojection by default, as it preserves datasets in their native projections to prevent unnecessary resampling and loss of spatial accuracy. General tutorials for reprojection of raster data are provided in the online documentation for users who require harmonised projections.”

And in the online documentation <https://bih-dmbs.github.io/CLUES/UserGuide/#data-enrichment>:

“After running the workflow, the required datasets will be downloaded and stored according to the general configuration specified in config/config.json. You can then enrich your data by linking environmental exposures to participant locations. Participant locations are expected to be provided in geographic latitude–longitude (WGS84). If location data are available in a different coordinate reference system, you must reproject them to WGS84 prior to linkage. A general tutorial on reprojection is available here: <https://py.geocompx.org/06-reproj>.”

4. I understand that for exposome type of studies, you just need to get the exposition variable values under the location of the individual data, so projection would not matter too much in this case. However, because you are stating that CLUES can be used in many other settings, how data are projected could matter greatly. If the projected data sets are used for zonal statistics over large areas, or for geographical accessibility modeling, having an equal area metric projection would be very important, for example. I would like to see some more details on this.

We agree with the reviewer that while projection choice is of minor importance for point-based exposome linkage, it becomes critical for analyses that operate on spatial areas, such as zonal statistics, polygon-based aggregation, or geographical accessibility modelling.

For example, computing zonal statistics or population-weighted averages over administrative regions (e.g., NUTS3) in a geographic CRS such as WGS84 may lead to systematic under- or overestimation of area-dependent quantities, with distortions that can reach approximately 10% depending on latitude and polygon geometry. In

such cases, reprojecting data to an appropriate equal-area CRS is recommended prior to analysis.

CLUES does not enforce a single projection for all downstream analyses, as the optimal equal-area CRS depends on the geographic region and analytical objective. Instead, CLUES provides guidance and example scripts for projection-sensitive workflows. In particular, the online documentation includes a dedicated section on [Enrichment for geographic areas \(https://bih-dmbs.github.io/CLUES/UserGuide/#enrichment-for-geographic-areas\)](https://bih-dmbs.github.io/CLUES/UserGuide/#enrichment-for-geographic-areas), which demonstrates how to perform polygon-based enrichment and zonal statistics.

5. My comments actually relate broadly to the box "Data harmonization" in Fig 3, because it's not very explicit how exactly datasets are harmonized. Add a sentence to clarify at which part of the workflow the harmonisation takes place.

We revised the Workflow specifications section and added a paragraph following Figure 2 that clarifies what data harmonisation entails in CLUES (lines 410–444):

“The CLUES framework is developed using the Snakemake workflow management system and made publicly available as open-source software (Figure 2). Its modular design allows for easy customization and regular updates. Configuration files define key parameters, such as selected variables, data storage locations, and the geographic and temporal extent, ensuring reproducible database creation. The Snakemake workflow coordinates dataset retrieval and processing, after which the harmonised outputs are stored in a structured geospatial database.

Figure 2 ...

In CLUES, data harmonisation ensures that all environmental layers conform to a consistent spatial extent, temporal coverage, and organisational structure. During execution, the source-specific Python download and processing scripts apply the harmonisation settings by retrieving the required spatial and temporal subsets and stitching tiled products into seamless rasters where necessary. Most datasets integrated in CLUES are natively provided in WGS84. MODIS vegetation indices (NDVI, EVI) are an exception, as they are distributed in a sinusoidal equal-area projection. All datasets are retained in their original CRS.

The workflow standardises directory organisation, naming conventions, and output formats (GeoTIFF, netCDF, CSV), ensuring that heterogeneous datasets can be queried consistently throughout the pipeline. During the subsequent linkage step, CLUES transforms participant coordinates, provided in geographic latitude-longitude, into the native CRS of each raster before extracting environmental values, enabling accurate point-level enrichment without resampling large raster datasets.”

6. I gather that you are not resampling raster data set within the workflow, but this is not really explicit and this could be mentioned somewhere.

We have now made this explicit in the “Workflow specifications” section (lines 408–447)), where we clarify that CLUES avoids unnecessary resampling and keeps rasters in their native projection.

8. Line 145-146: how does the CLUES facilitate the linkage with geolocation, especially when you have residential address on one side (individual), and coordinates on the other (exposure variables)? Also, you mention "routes". What do you mean exactly? Tracked individual's data along which you want the exposure values? Specify

We would like to clarify that CLUES performs the linkage between individual-level data and environmental exposures by working with geolocation inputs. These can be coordinates provided directly, residential addresses that users convert to coordinates, or routes (movement trajectories) supplied by users as sequences of coordinates.

To support users who start from address-based data, we have added a geocoding script to the online documentation (<https://github.com/BIH-DMBS/CLUES/tree/main/scripts/address2coord>), which converts residential addresses into latitude/longitude before linkage, allowing users to prepare their data without requiring external GIS tools.

Regarding *routes*, we clarify that CLUES allows users to supply any predefined trajectory (e.g., commuting paths, GPS traces, walking routes). The framework will then extract environmental exposures along those paths. We have also added a route processing example notebook (https://github.com/BIH-DMBS/CLUES/blob/main/notebooks/linkage_of_tracks_2_netcdf_geoTiff.ipynb) in the online documentation to assist users.

We have revised the manuscript (lines 155–161) as follows:

“CLUES links individual-level data to environmental exposures using geographic coordinates as the required input (Figure 1). Coordinates may be provided directly or obtained from address information using the geocoding script included in the online documentation. Routes (e.g., movement paths or GPS traces), which consist of coordinate sequences, can be linked through the standard workflow. A route-processing example notebook is also available online.”

The online User Guide (<https://bih-dmbs.github.io/CLUES/UserGuide/#data-enrichment>) has also been updated as:

“If your dataset contains address-based locations rather than coordinates, a geocoding script is provided [here](

DMBS/CLUES/tree/main/scripts/address2coord) to convert address lists into latitude–longitude pairs.

For users working with routes or movement paths (e.g., commuting trajectories or GPS traces), no special linkage procedure is required: as long as the route is represented as a series of coordinate points, it can be processed using the standard CLUES enrichment workflow. An example notebook illustrating how such route-based exposures can be linked and visualised after linkage is provided [here](https://github.com/BIH-DMBS/CLUES/blob/main/notebooks/linkage_of_tracks_2_netcdf_geoTiff.ipynb).”

8. In the section "Usage policies", can you specify whether the "terms of use" of each data set are downloaded as well and readily available to the user. If they are not, that would be very cumbersome for the user to get them individually! Also, it would be great if an AI tool could correctly summarize all terms of use of downloaded data sets to flag anything of interest for the user.

We have added the licence information and reference links for each primary data source to online documentation (<https://bih-dmbs.github.io/CLUES/dataCatalog/>). We have also updated the “Usage policies” section of the manuscript to clarify that every dataset integrated into CLUES is governed by its own terms of use and to indicate where this information can be found. The revised text (lines 730–734) reads:

“Usage policies

All datasets integrated into CLUES are open-source and publicly available. Each primary data source is subject to its own terms of use. Source-specific license information and reference links are listed in the online Data Catalogue. Users should review these usage policies prior to data use to ensure correct attribution and compliance.”

Reviewer #2 (Remarks on code availability):

I have not reviewed the code source itself, but I have reviewed the github repository and it seems complete with user guide, license, config files, etc.

Reviewer #3 (Remarks to the Author):

This manuscript proposed a framework to integrate geospatial datasets in biomedical research settings. They claim it is the first open-source tool to generate individual-level exposure data, while it's not necessary to have any experience working with geospatial data. I believe that the proposed framework is a nice idea and could be very beneficial to public health research, but some concerns need to be addressed first.

1. CLUES currently requires users to have a working knowledge of Python and programming fundamentals. Have you considered developing an R implementation, given R's popularity in spatial statistics and its relatively user-friendly interface for statistical users

We fully acknowledge R's strong ecosystem for spatial statistics and its popularity among researchers. However, several components of the workflow, particularly large-scale data retrieval, API-based downloading, and parallel raster processing, rely on Python's geospatial stack and its mature integration with workflow managers such as Snakemake and with HPC environments. These features were essential for handling large datasets and configuration-driven processing at continental and global scales. Some of the data sources used in CLUES (e.g., Copernicus Climate Data Store) provide fully supported client libraries only in Python. Although these APIs are technically language-agnostic, reproducing the same functionality in R would require substantial custom implementation for authentication, request handling, pagination, retries, and large-batch downloads.

Developing and maintaining a full parallel R implementation with equivalent functionality and performance would therefore involve significant duplication of effort and risk divergence between versions. That said, only basic Python knowledge is required to run the workflow, as CLUES is fully scripted and users typically interact with it via editable configuration files and predefined commands.

In addition, all generated environmental datasets (netCDF, GeoTIFF, CSV/geometry) can be seamlessly imported into R and used with standard spatial and statistical workflows. To provide an R-based example, we have added a notebook in the online documentation demonstrating how to load and visualise netCDF files generated via CLUES (https://github.com/BIH-DMBS/CLUES/blob/main/notebooks/R/r_demo.ipynb).

1a. Or alternatively, a graphical user interface (GUI) to enhance accessibility for users with limited programming experience?

We agree that a GUI can improve accessibility for users with limited programming experience. At this stage, however, CLUES is designed primarily as a workflow engine for geospatial processing and automated data integration, where reproducibility, transparency, and version control are essential. Developing and maintaining a GUI of comparable functionality would require substantial resources and introduce additional maintenance complexity, particularly given the range of supported datasets and the need to manage long-running operations, authentication, and workflow monitoring.

Instead, we have focused on lowering the barrier to entry through clear configuration-file templates, step-by-step user guides, example notebooks, and a JupyterHub deployment with preconfigured workflows. Therefore, a GUI is outside the current

project

scope.

2. The description suggests that CLUES applies mean and standard deviation filters using circular kernels of varying sizes to capture local and broader context. Could you clarify whether these kernels represent multiple fixed spatial scales or if an adaptive (variable-width) bandwidth approach is used? If fixed, have you considered implementing adaptive kernels to better account for spatial heterogeneity in data density?

Thank you for this insightful question. Indeed, in CLUES we apply mean and standard deviation filters using circular kernels of varying sizes to capture both local and broader spatial context. The specific kernel sizes are user-defined and can be configured in the CLUES configuration file. Following the reviewer's suggestion, we investigated adaptive (variable-bandwidth) kernel approaches. These methods, commonly used in spatial statistics and especially in adaptive kernel density estimation, are designed for datasets with uneven point densities, where the kernel radius expands in sparse regions and shrinks in dense regions. However, CLUES operates on continuous raster surfaces rather than irregular point patterns, and raster data do not exhibit the type of spatial sparsity or density variability for which adaptive kernels provide the greatest benefit. As an alternative, computing local neighborhoods during the linking step can also capture local variability while sparing computational resources.

For applications where users wish to apply adaptive kernels to external datasets before using CLUES, we have included a stand-alone script that applies adaptive kernels to GeoTIFF rasters (https://github.com/BIH-DMBS/CLUES/tree/main/scripts/adaptive_kernel_filter).

We have added this information to the manuscript (lines: 518–523):

“CLUES applies fixed-radius circular kernels, with kernel sizes defined by the user in the configuration file. Adaptive bandwidth kernels ⁴⁰ are not part of the core workflow, as CLUES operates on continuous raster surfaces rather than irregular point patterns, where adaptive approaches typically offer the greatest benefit. However, an optional stand-alone script is provided in the online documentation for users who wish to apply adaptive smoothing to external datasets.”

3. You mention that socioeconomic data are provided as CSV files. Could you elaborate on the decision to use CSV rather than a geospatially compatible tabular format such as DBF or GeoPackage? CSVs can lack important metadata, field type definitions, and direct compatibility with shapefiles, which may introduce issues with field name truncation or type conversion during import. A DBF or similar format could improve interoperability with common GIS software and preserve data integrity.

The socioeconomic data incorporated in CLUES originate from the ESPON repository, which distributes these datasets natively as CSV files together with a single accompanying geometry.csv file containing spatial boundaries in WKT format. CLUES

preserves this structure to ensure fidelity to the original source, minimise duplication of large geometry objects, and maintain compatibility with ESPON's API-based retrieval system.

CSV also offers practical advantages such as universal readability, compatibility with scripting-based analytical workflows, and row-level metadata that makes each file self-describing, while avoiding limitations of DBF/shapefiles (e.g., truncated field names, limited data types, encoding issues).

To address the reviewer's suggestion about GIS interoperability, we have added a page in the github documentation on data formats and have added a conversion script that merges indicator CSVs with the geometry file and exports a GIS-ready GeoPackage and a prototype metadata.json, preserving the advantages of CSV while enabling full GIS compatibility and reproducibility (https://bih-dmbs.github.io/CLUES/data_formats/).

We have added a short text in the manuscript (lines: 536–539) as: “Additional information on data formats, together with utility scripts to optionally convert socioeconomic CSV indicators into GIS-compatible formats (e.g., GeoPackage), is available in the online documentation (https://bih-dmbs.github.io/CLUES/data_formats/).”

4. Could the authors clarify how easily CLUES can be adapted to other regions, such as the U.S. or low- and middle-income countries, where equivalent data infrastructures and APIs may differ or be unavailable? What level of customization would be required for non-European applications?

CLUES was built with a modular, data-agnostic and region-agnostic architecture intended for use at a global scale. The workflow does not assume any specific region context. Instead, each dataset, whether global, national, or local, is defined through a configuration file that specifies how it should be accessed and processed. Adapting CLUES to regions outside Europe does not require modifying the core codebase but simply adjusting the general workflow configuration file.

Most environmental and raster-based components used in CLUES, such as digital elevation, land cover classifications, climate and air-pollution grids, and satellite-derived surface variables, are globally available. These layers are therefore highly portable across continents.

In contrast, components that rely on region-specific socioeconomic data or administrative boundaries depend on the availability and quality of local data. For example, ESPON repository provides a harmonised framework for European socioeconomic indicators. Outside Europe, users may need to source national datasets and perform additional harmonisation (especially when combining several regions) or metadata curation. CLUES supports the integration of additional datasets by design and provides an applied tutorial in the online documentation for users who wish to perform such extensions themselves (<https://bih->

dmbs.github.io/CLUES/UserGuide/#adding-a-new-dataset-to-clues-worldpop-example).

We benchmarked CLUES performance using three countries with different spatial extent, such as France (1,247,000 km²), Nepal (362,000 km²), and Suriname (210,000 km²), to demonstrate scalability across diverse geographic sizes and global contexts. We have incorporated the results of these evaluations in the Supplementary Information (SI1 Scalability and Performance evaluation).

This information has been added into the manuscript (lines: 722–729):

“Beyond the default data catalogue, additional datasets can be integrated as scientific needs and available data evolve. CLUES is actively extended with new sources, and users may incorporate datasets of their choice by following the dedicated tutorial to the online User Guide. The tutorial demonstrates how to create a source-specific configuration file, register the new source within the Snakemake workflow, and implement the corresponding download function.”

In practice, CLUES has already been applied in projects for Norway, Germany, the UK, France, and China, and we have conducted test runs for Nepal and Suriname, demonstrating the workflow’s adaptability. Successful deployment in new contexts depends on the availability of comparable environmental, socioeconomic, and administrative data, as well as the effort invested in data wrangling, harmonization, and metadata management. Where high-quality, well-documented data exist, adaptation is relatively straightforward. Where data are sparse or fragmented, customization becomes more challenging but remains feasible with appropriate data engineering and documentation.

Manuscript changes (lines: 218–224):

“The workflow is not tied to any specific geographic region. Most default environmental datasets have global coverage, and researchers can tailor the environmental database to their research needs by defining the geographic extent, time window, and relevant environmental variables through editable configuration files. The framework’s modularity also supports the integration of additional datasets when available, enabling seamless scaling across cohorts, regions, and research questions.”

5. Given that CLUES harmonizes coordinate reference systems and bounding boxes across diverse sources, how does the workflow handle discrepancies in datum definitions (e.g., WGS84 vs. ETRS89) and projection distortions at global scales? Has this been tested outside of Europe?

Most datasets integrated in CLUES are natively provided in WGS84, including all temporally resolved datasets, meaning that no datum transformation (e.g., between WGS84 and ETRS89) is required. As a result, discrepancies in datum definitions do not arise in the default workflow.

CLUES does not enforce a single projection for all datasets. Instead, it harmonises coordinate systems at the linkage stage. This approach preserves the native spatial accuracy of datasets such as MODIS NDVI/EVI while ensuring consistent and correct linkage across heterogeneous projections. This has been clarified in the manuscript (lines 435–444):

“... Most datasets integrated in CLUES are natively provided in WGS84. MODIS vegetation indices (NDVI, EVI) are an exception, as they are distributed in a sinusoidal equal-area projection. All datasets are retained in their original CRS.

.... During the subsequent linkage step, CLUES transforms participant coordinates, provided in geographic latitude-longitude, into the native CRS of each raster before extracting environmental values, enabling accurate point-level enrichment without resampling large raster datasets.”

6. The privacy-preserving linkage approach is commendable. However, could the authors elaborate on how CLUES ensures compliance with non-EU data protection frameworks (e.g., HIPAA in the U.S.)? Are there built-in options for differential privacy or coordinate perturbation beyond rounding or masking?

CLUES is designed so that all privacy-sensitive operations, particularly in regard to exposure-linkage, are performed locally by the authorised data controller, and no personal geolocation data leave the institutional environment. Therefore, CLUES itself does not enforce a specific legal framework, such as GDPR, or HIPAA, but enables compliance by ensuring that sensitive data are never transferred to external servers and the linkage can occur entirely within a secure computing environment. For the straightforward approach of blurring we have added a script (https://bih-dmbs.github.io/CLUES/Scripts_Notebooks/).

CLUES facilitates the non-mandatory implementation of exposure linkage, and is compatible with regulatory contexts because:

- All geocoding and environmental linkage run locally, so no geolocation data are transmitted to third-party APIs or external cloud services
- Users can enhance privacy through practical techniques such as: adding random offsets to coordinates (jitter/blurring), aggregating or snapping locations to grid cells, generating multiple plausible locations per participant, spatial smoothing, or applying geo-indistinguishability for formally quantifiable differential privacy. However, CLUES does not include built-in formal differential privacy mechanisms.
- CLUES does not overwrite or store original coordinates

We have addressed these points in the manuscript (lines: 250–259):

“Moreover, CLUES supports data handling with respect to applicable data protection regulations by enabling secure, local processing of sensitive geolocation data. This ensures that linkage between individual-level data and environmental exposures can be performed entirely within authorized institutional environments, eliminating the need for external data transfer and reducing privacy risk. Users may optionally apply additional privacy-preserving methods before linkage, such as adding small random offsets (jittering), aggregating or snapping coordinates to grid cells, generating multiple plausible locations, or applying spatial smoothing, when required by local governance policies. These methods are not built into CLUES and should be applied externally. An example script illustrating coordinate jittering is provided in the online documentation.”

Reviewer #3 (Remarks on code availability):

The github and code provided is very reproducible for those who have programming experience in Python. The tutorials are also great to get started and follow along with some of their examples in Europe. All important documentation are included, such as README files, instructions, and comments throughout scripts.